# Artificial Intelligence and Child Abuse and Neglect: A Systematic Review

**DOI:** 10.3390/children10101659

**Published:** 2023-10-06

**Authors:** Francesco Lupariello, Luca Sussetto, Sara Di Trani, Giancarlo Di Vella

**Affiliations:** Dipartimento di Scienze della Sanità Pubblica e Pediatriche, Sezione di Medicina Legale, Università degli Studi di Torino, 10126 Torino, Italy

**Keywords:** artificial intelligence, child abuse, machine learning, deep learning, neglect, child sexual abuse, violence against children

## Abstract

All societies should carefully address the child abuse and neglect phenomenon due to its acute and chronic sequelae. Even if artificial intelligence (AI) implementation in this field could be helpful, the state of the art of this implementation is not known. No studies have comprehensively reviewed the types of AI models that have been developed/validated. Furthermore, no indications about the risk of bias in these studies are available. For these reasons, the authors conducted a systematic review of the PubMed database to answer the following questions: “what is the state of the art about the development and/or validation of AI predictive models useful to contrast child abuse and neglect phenomenon?”; “which is the risk of bias of the included articles?”. The inclusion criteria were: articles written in English and dated from January 1985 to 31 March 2023; publications that used a medical and/or protective service dataset to develop and/or validate AI prediction models. The reviewers screened 413 articles. Among them, seven papers were included. Their analysis showed that: the types of input data were heterogeneous; artificial neural networks, convolutional neural networks, and natural language processing were used; the datasets had a median size of 2600 cases; the risk of bias was high for all studies. The results of the review pointed out that the implementation of AI in the child abuse and neglect field lagged compared to other medical fields. Furthermore, the evaluation of the risk of bias suggested that future studies should provide an appropriate choice of sample size, validation, and management of overfitting, optimism, and missing data.

## 1. Introduction

Epidemiologic analyses of child abuse and neglect have revealed that many children are abuse victims worldwide. According to the Centers for Disease Control and Prevention, “at least 1 in 7 children have experienced child abuse and neglect in the past year in the United States” [1]. All societies should address the issue of child abuse and neglect, as it can lead to a range of acute and chronic consequences. These include the intergenerational transmission of child maltreatment, low educational achievements, limited employment opportunities, mental health problems, attempted suicide, alcohol and drug addiction, obesity, chronic pain in adulthood, and more. It is crucial to take a careful approach to dealing with this issue to prevent these harmful outcomes [2,3,4].

As one of the most critical goals consists of a timely identification/diagnosis of children at risk or who are victims of abusive acts, many researchers have tried to find new predictive tools to counteract the abovementioned phenomenon.

Many researchers have developed new predictive artificial intelligence (AI) tools to assist healthcare professionals in daily activities. In some medical fields, these tools are already well established, being widely diffused in practical routines [5]. Indeed, several studies have demonstrated the high accuracy of AI models in diagnosis, prediction of the clinical course of a disease, assessment of therapeutic interventions, etc. [6,7].

The state of the art of implementing AI tools in child abuse and neglect is unknown. No studies have comprehensively reviewed the types of AI models that have been developed/validated. Furthermore, no indications about the risk of bias in these studies are available. For these reasons, we conducted a systematic review of the PubMed database to answer the following questions: “which is the state of the art about the development and/or validation of AI predictive models useful to contrast child abuse and neglect phenomenon?”; “which is the risk of bias of the included articles?”.

## 2. Material and Methods

The study followed the Preferred Reporting Items for Systematic Reviews and Meta-Analyses (PRISMA) indications [8]. The authors carried out a literature search in the PubMed database in March 2023. The following syntax was used: (“Machine Learning” [Mesh] OR “Algorithms” [Mesh] OR “Artificial Intelligence” [Mesh] OR “Diagnosis, Computer-Assisted” [Mesh] OR “Therapy, Computer-Assisted” [Mesh] OR “Deep Learning” [Mesh] OR “Big Data*” [Mesh] OR “artificial intelligen*” [tw] OR “machine learn*” [tw] OR “algorithm*” [tw] OR “computer assist*” [tw] OR “deep learn*” [tw] OR “Big Data*” [tw]) AND (“Child Abuse” [Mesh] OR “child maltreatment*” [tw] OR “child violence*” [tw] OR “child abus*” [tw] OR “child physical abus*” [tw] OR “violence against child*” [tw] OR “child punish*” [tw] OR “child exploitat*” [tw]).

Three reviewers (LS, SD, and FL) independently evaluated the titles and the abstracts. The full-text articles considered eligible by the three reviewers were screened together. The inclusion criteria were: articles written in English and dated from January 1985 to 31 March 2023; publications that used a medical and/or protective service data dataset to develop and/or validate AI prediction models.

A quantitative synthesis was not appropriate because of the heterogeneity of the present review. The authors provided a qualitative synthesis of the results using a narrative approach. The data extracted from the included articles were: authors, year of publication, input data, age/sex, type of abuse, output/prediction, AI model, dataset’s size, accuracy, sensitivity, specificity.

The appropriate evaluation of the performance and/or robustness of the AI model of the reviewed articles is controversial. For this reason, we chose accuracy, sensitivity, and specificity because many studies have used them.

The risk of bias was assessed using the prediction model risk of bias assessment tool (PROBAST) [9]. The PROBAST tool classifies the risk of bias as low, moderate, or high, evaluating twenty signaling questions from the following four domains: participants, predictors, outcomes, analyses [10]. For example, this tool proposes the following questions for the analysis domain: “4.1 Were there a reasonable number of participants with the outcome?”; “4.2 Were continuous and categorical predictors handled appropriately?”; “4.3 Were all enrolled participants included in the analysis?”; “4.4 Were participants with missing data handled appropriately?”; “4.5 Was selection of predictors based on univariable analysis avoided?”; “4.6 Were complexities in the data accounted for appropriately?”; “4.7 Were relevant model performance measures evaluated appropriately?”; “4.8 Were model overfitting and optimism in model performance accounted for?”; “4.9 Do predictors and their assigned weights in the final model correspond to the results from the reported multivariable analysis?”. “If the answer to any of the signaling question is ‘no’ or ‘probably no’, there is a potential for bias” [10]. For the other three domains (participants, predictors, and outcome), the PROBAST tool identifies other signaling questions; if the answers are “no” or “probably no”, then there is a potential for bias. Finally, if ≥1 domain is judged to be at a high risk of bias, then the PROBAST tool will define the study as being “with high risk of bias” [10].

## 3. Results

The reviewers screened 413 articles. Eleven were excluded because they were not in English. Then, 330 records were excluded from the title and abstract. The remaining 72 articles’ full-text reading was used for the final inclusion/exclusion process. A total of 65 full-text articles were excluded. Thus, 7 studies were included in the systematic review (Figure 1) [11,12,13,14,15,16,17]. In Table 1, the authors summarize the following indexes: authors, year of publication, input data, age/sex, type of abuse, output/prediction, AI model, dataset’s size, accuracy, sensitivity, specificity. As shown in Table 1, the included publications were gathered in three years (2020–2022), except for one article published in 2000.

The types of input data were heterogeneous: radiologic imaging in one case, demographic and clinical characteristics in two cases, text of medical records in two cases, self-figure drawing in one case, child protection system data in one case.

The age of the children was stated in four studies: two of them were focused on infants (≤1 year old), one on people ≤ 18 years old, and one on children aged 5–17 years.

In five studies, no specific indications about the sex of the examined children were reported; in the remaining two studies, children of both sexes were analyzed.

In four studies, the authors reported that the analyzed cases were indicative of physical abuse; two manuscripts analyzed child sexual abuse cases; one article focused on multiple forms of abuse.

In six cases, the output was the differentiation between abused and not-abused children; in one study, the target was to predict the development of major depressive disorder versus post-traumatic stress disorder after child sexual abuse.

The authors of the seven reviewed articles chose the following AI algorithms: artificial neural networks, convolutional neural networks, natural language processing. These algorithms were used to process datasets with a median size of 2600 cases. In three studies, the datasets contained more than 1000 cases; two studies analyzed datasets with less than 200 cases.

Regarding the evaluation of AI performance, accuracy was used in six studies, sensitivity in five, and specificity in four. A quantitative synthesis of these data was not carried out due to the heterogeneity of the reviewed articles. Data about accuracy, sensitivity, and specificity are depicted in Table 1.

A summary of the risk of bias for each model is reported in Table 2. The PROBAST tool analysis pointed out that all seven studies had a high risk of bias. In particular, the analysis domain of all seven manuscripts was classified as having a high risk of bias because of the low number of cases (4.1), the absence of appropriate indications about missing data (4.4), and the lack of accounting for model overfitting, underfitting, and optimisms (4.8). In addition, most studies were limited by the absence of external/internal validation.

## 4. Discussion

### 4.1. General Considerations

In the past decade, there has been a dramatic increase in AI implementation in medicine. Many authors have suggested using AI algorithms to facilitate and automate decisions in healthcare settings [10]. AI has demonstrated excellent accuracy in diagnosing skin cancer, intracranial lesions, retinal modifications, etc. Moreover, AI algorithms can predict the clinical progression of medical conditions [5,6,7].

Despite the many possible applications of these tools, this review pointed out that a few studies have aimed to counteract the child abuse and neglect phenomenon by developing/validating AI predictive algorithms. Only 7 manuscripts were included after a careful analysis of 413 results. The research in this field lags compared to that in other medical fields.

The strength of this review is that we could point out the state of AI research regarding child abuse as follows.

#### 4.1.1. Age, Sex, and Type of Abuse

The literature analysis highlighted that the included studies did not cover all age ranges: no studies focused on children aged between 2 and 5 years. Moreover, the children’s ages were highly heterogeneous (#1 and #4 articles studied infants; #2 article included people aged ≤ 18 years; #6 article analyzed children aged 5 to 17 years). In addition, the authors did not clearly specify the ages of their samples in three articles (#3, #5, #7). Similarly, the sex of the children was not clearly stated in most of the seven manuscripts (#1, #3, #4, #5, #7).

Regarding the type of abuse, #1, #2, #3, and #4 studies developed AI models to predict physical abuse. On the contrary, #5 and #6 manuscripts focused on child sexual abuse. The decision to choose these topics is undoubtedly justified by the scientific literature data highlighting the need to identify early signs of physical maltreatment and sexual abuse due to their adverse acute and chronic sequelae [1,2,18]. However, it is surprising that until now, articles specifically focused on neglect are not available. In fact, epidemiologic analyses have reported that neglect is the most common form of abuse in children (with high morbidity and mortality), but it is usually hard to identify [19]. For this reason, developing AI models—capable of assisting healthcare and/or social service professionals facing this type of abuse—could significantly help predict the risk and/or identify the neglect.

Since this review highlighted that the research on child abuse and neglect and AI predictive models is at an early stage, the data above—that demonstrate a lack of covering ages, sex distribution, and type of abuse—are not unexpected. However, according to the scientific literature, the child abuse and neglect phenomenon can profoundly change depending on the specific category of these three variables [1,20,21]. Thus, before creating AI predictive algorithms, it could be helpful to conduct studies evaluating how data about age, sex, and type of abuse should be managed in AI developing processes.

#### 4.1.2. Input Data and Types of AI Models

Every day, several medical images are produced during standard medical routines. These images come from endoscopy, pathology, ultrasound, radiology (computed tomography—CT, magnetic resonance imaging—MRI, X-ray, and positron emission-computed tomography—PET), etc. Healthcare professionals usually spend many hours per day analyzing and classifying these images for diagnostic and/or therapeutic purposes [22]. To facilitate these operations, many researchers have designed AI tools capable of giving specific predictions in the case of medical images.

Object detection and image classification are two of the most common applications of AI. Convolutional neural networks (CNNs) are frequently used in this field. A CNN is a particular form of machine learning (deep learning) that simulates the structure of the human brain. The CNN is capable of image classification, semantic segmentation, and object detection [23,24]. For these reasons, CNNs have been widely employed by many researchers to develop AI algorithms useful in medical practice for detection and imaging classification. For example, many manuscripts have reported the implementation of CNNs for detecting and classifying neoplasms by analyzing CT, MRI, and PET images in oncology [25].

That said, object detection and image classification using CNNs are promising tools for child abuse and neglect because they could assist physicians in analyzing medical images of child abuse. For example, CNN tools could help radiologists differentiate between accidental and abusive bone fractures. Despite the latter promising application, this review highlighted only one article (#1) in which radiologic images are used. Using a residual neural network, Tsai and Kleinman (#1 article) proposed a pilot study to differentiate distal tibial classic metaphyseal lesions [11]. They stated that their preliminary research “should stimulate further efforts to leverage the power of this emerging technology as a complementary tool to increase the diagnostic accuracy and confidence in the radiologic evaluation of suspected infant abuse” [11].

Possible applications of CNNs are numerous in this field. This consideration is confirmed by the #5 article in which Kissos et al. created a CNN tool to differentiate self-drawings indicative or not indicative of child sexual abuse [15]. The authors stated that “these preliminary results suggest that CNN, when further developed, can contribute to the detection of child sexual abuse” [15].

In medical practice, another possible application of AI models relies on analyzing the free text of medical records through the so-called natural language processing (NLP). NLPs can infer meaning from words by analyzing text and speech [26]. For example, NLP implementation gave birth to several studies in which text from medical records was used to assist physicians in predicting and/or diagnosing several gastrointestinal diseases (such as hepatobiliary/pancreaticobiliary pathologies and inflammatory bowel diseases) [27]. In three manuscripts (#2–#4), the authors tried to develop/validate NLP models for free text. Since the types of records were highly heterogeneous, a proper comparison between these manuscripts was impossible. However, the aforementioned authors demonstrated that NLP could help predict the risk of abuse in children by processing medical records and/or social services reports.

### 4.2. Considerations about the Risk of Bias—PROBAST Tool

The analysis of the seven included articles using the PROBAST tool pointed out that the risk of bias was high for all seven articles (Table 2). The main issues were related to the methodologies used to analyze the data. These issues are discussed in detail as follows.

#### 4.2.1. Number of Participants with the Outcome

According to PROBAST indications about analysis methods, a large sample size usually leads to more precise results in medical research. However, for studies on prediction models, it is also essential to consider the number of participants with the outcome [9,10]. In the case of a binary outcome, “the effective sample size is the smaller of the 2 outcome frequencies, ‘with the outcome’ and ‘without the outcome’” [10]. For example, suppose the prediction model aims to distinguish between abused children (with the outcome) and not-abused children (without the outcome) and the analyzed database is composed of 400 abused children and 600 not-abused children. In that case, the effective sample size will be 400 (the smaller of the two outcome frequencies).

The performance of a prediction model tends to be overestimated if the aforementioned sample size is small. Historically, this issue has been addressed by many authors who have postulated that the effective sample size should be chosen following the so-called number of events per variable (EPV). The EPV is calculated by dividing the number of events by the number of predictor variables considered in developing the prediction model. For example, if the prediction model aims to distinguish between abused children (with the outcome) and not-abused children (without the outcome), the analyzed database is composed of 400 abused children and 600 not-abused children, the effective sample size is 400 (the smaller of the 2 outcome frequencies), and the predictor variables chosen for the model are 40. In this case, the EPV will be 10 (400/40 = 10). Even if an EPV of at least 10 is usually accepted to minimize overfitting, recently, some authors have suggested an EPV of at least 20 [10,28,29]. In addition, the scientific literature has reported that “prediction models developed using machine learning techniques often require substantially higher EPVs (often > 200) to minimize overfitting” [10].

That said, it is essential to note that, in the seven included articles, no authors specifically calculated the EPV to evaluate if the sample size was large enough to reduce the overfitting of the AI tool. Even if all seven studies were carried out using machine learning techniques, the authors did not provide helpful data demonstrating that the dataset size was large enough (EPV > 200) to sustain the prediction model statistically. Moreover, the datasets comprised less than 200 cases in #1 and #6 articles.

It is important to note that the authors of #1, #2, #5, and #6 articles demonstrated an awareness of the limitation of their analyses due to the small sample size. Indeed, they all reported that further studies with larger sample sizes are needed. In particular, Kissos et al. (#5 article) stated that “further research is needed to achieve satisfactory CSA CNN prediction levels. To do so, big data (thousands of self-figure drawings) from different cultures and ethnic groups across different ages are needed” [15].

In other fields, AI algorithms already used in clinical routine are based on larger datasets. For example, Esteva et al. developed a deep neural network tool to classify skin cancer using 129,450 clinical images [5]. Gulshan et al. created and validated a deep-learning algorithm for detecting diabetic retinopathy, analyzing 128,175 retinal images [6]. Chilamkurthy et al. proposed an AI algorithm for detecting clinical findings in head imaging using 313,318 images [7]. Compared to these studies, the ones included in the present review were carried out with smaller samples and without specific indications about the EPV. This negatively affects the impact of their results.

#### 4.2.2. Validation

Even if external validation for the prediction model is not mandatory, the study can highlight possible interventions to improve it. When external validation is carried out, the development of the model is followed by quantifying its predictive performance using data external to the development sample. External validation was carried out in one included manuscript (#4 article). In fact, Tiyyagura et al. (#4 article) developed their predictive AI model using the cases observed from 10 October 2019 to 1 January 2020 and validated the same model using the data observed in the same hospital sites from 1 September 2020 to 15 December 2020. This approach is a correct form of external validation (called temporal validation) because it tests the predictive AI algorithm on a sample collected in a later period [10].

It is important to note that “randomly splitting a single data set into a development and a validation data set is often erroneously referred to as a form of external validation but actually is an inefficient form of internal validation, because the 2 data sets created in this way differ only by chance and the sample size of model development is reduced” [10]. The latter approach was carried out in the #2, #3, #5, and #6 articles, in which the population was randomly divided into training and testing samples. Thus, their results about the performance of their AI models could have been negatively affected by the lack of a proper validation process.

The external validation of AI algorithms is usually more accessible when the research is multicentric. In fact, in these cases, the researchers can develop the predictive model using a large dataset and then validate it using another dataset created in another center. For example, Chilamurthy et al. decided to validate a deep learning algorithm to detect critical findings in computed tomography scans using an additional validation dataset collected from centers different from those used in the development phase [7].

On the contrary, most of the included articles were not designed as a multicentric analysis, thereby not allowing this helpful approach.

#### 4.2.3. Overfitting and Optimism

The systematic review highlighted that a discussion of overfitting and optimism in the AI model performance was not carried out in most of the seven included articles.

The literature reports that some authors can use inadequate methods to identify and/or correct the optimism of AI models. Some researchers tend to split the same dataset using one sample for development and the other for testing. This approach is an incorrect way to measure optimism (bootstrapping and cross-validation should be preferred) [10].

That said, the authors of #2, #3, #5, and #6 articles decided to use the aforementioned approach, quantifying the performance of their AI models using the same dataset from which the algorithms were created. Nevertheless, this approach (apparent performance) gives optimistic estimates due to the so-called overfitting (i.e., the data used for the development and the model are too adapted). Moreover, the optimism is larger if few outcome events are available (small population) and the EPV is too small.

#### 4.2.4. Participants with Missing Data

The scientific literature suggests that—when it comes to creating a predictive model—participants with missing data should be carefully managed. Indeed, simply excluding these participants from the analysis causes errors because the analyzed sample is, therefore, not selected randomly. The correct handling of individuals with missing data requires using specific methods such as the so-called multiple imputation [30,31,32,33]. The present review highlighted significant issues about this topic. Indeed, the authors did not report a specific disclosure about missing data in #1, #4, #5, and #6 articles. When a manuscript does not provide a disclosure, this usually means that these data have been omitted from the analysis because statistical software automatically excludes individuals with missing values. Moreover, in the #3 article, the authors stated that “Records with < 2 notes remaining were excluded” [13]. Marshall et al. (#7 article) decided to address this item by removing “referrals with large amounts of missing data” [17]. In light of the above, the methodology used in these two articles (#3 and #7) appears not to be consistent with the indication of the scientific literature.

### 4.3. Ethical and Social Implications

It is well established that incidents of child abuse are significantly underreported and underdiagnosed due to the inability to identify them. Children who have been abused may seek medical assistance, but the abuse can go undetected, leading to further harm, injuries, and even death. Recognizing and diagnosing child abuse at an early stage significantly reduces the risk of recurrence and improves outcomes for the child. It is crucial to acknowledge demonstrated findings of child abuse when they occur. Misinterpreting markers of abuse as normal or as attributable to another diagnosis (false negative) can lead to the child returning to an unsafe environment where they may experience further horrific injury [34,35,36,37].

On the one hand, the scientific community is also aware that false positives can have harmful consequences. Incorrectly diagnosing child abuse can lead to the child not receiving the appropriate care, resulting in legal issues in civil, juvenile, family, divorce, and criminal courts. It can also lead to the wrongful removal of a child from their home and an innocent individual being falsely accused, convicted, and incarcerated. These outcomes can cause immeasurable harm to the children, individuals, and families involved [34,35,36,37].

It is often overlooked that balancing the risk of a false negative with that of a false positive is a fundamental ethical calculation. Healthcare professionals may be called to choose which outcome (false positive or false negative) is more critical to avoid. To better understand the latter sentence, it can be helpful to consider what happens in another medical field. The United States Preventive Services Task Force made recommendations against routine screening mammography for women aged 40–50. The task force chose to prioritize the harms of false positives (i.e., overdiagnosis, unnecessary anxiety, costs, procedures, and morbidity) over the potential lives that could be saved through screening. This decision ultimately involves a values-based determination of which outcome has higher costs. Similar considerations can be related to the decisions to prioritize false positives or false negatives in child abuse cases.

That said, it is essential to acknowledge that AI tools (especially those based on machine learning techniques) have been described as a black box, and healthcare professionals can only control the input data. AI algorithms and their outputs are often shrouded in complex and opaque reasoning. When faced with such a nebulous black box, it becomes nearly impossible to comprehend or delineate how a machine arrives at its conclusions [38]. This presents a significant ethical issue in the child abuse and neglect field, as using these tools can lead to a lack of control over the inevitable balancing between false positives and false negatives, adding another layer of complexity to an already complex field.

Risk models based on machine learning are typically evaluated using various predictive performance metrics such as precision, recall, etc. However, even when a risk model with a 99% precision is developed, it is still prone to errors in 1% of cases. In reality, many clinical machine learning-based risk models do not achieve such high predictive performance. These errors can have significant social and emotional consequences, especially in the domain of child abuse and neglect, where families found to be suspicious of abuse and neglect could face severe outcomes [39]. Furthermore, the performance evaluation metrics for traditional machine learning-based risk models assume that false positives and false negatives carry equal weights. However, it has not been established or validated that this is true for child abuse and neglect cases [39].

To minimize the abovementioned issues, all researchers should always guarantee strict compliance with scientific indications (such as the ones proposed by the PROBAST tool) about how an AI predictive model should be created. This strict compliance should be mandatory to minimize the negative scenarios these predictive tools may generate if not appropriately designed. Indeed, the need to identify new predictive tools to counteract child abuse and neglect does not justify using shortcuts in developing AI algorithms. As suggested by this review, an appropriate choice of sample size, validation, and management of overfitting, optimism, and missing data should always be carried out.

The integration of AI in clinical practice has raised concerns about patient privacy and data sharing. AI systems rely on vast amounts of medical information to provide accurate and reliable responses. However, this concentration of data poses valid concerns about the security and privacy of protected healthcare information, given the potential for unauthorized breaches. Past incidents of unauthorized access to healthcare databases highlight this potential vulnerability. Patients may also be hesitant to consent to the use of private data for training AI, given the possibility of external systems accessing their information without their knowledge. As technology continues to advance, the healthcare industry must take steps to ensure patient data are safeguarded [40].

### 4.4. Future Perspectives

In recent years, AI has made remarkable progress in perception, which is the process of interpreting sensory information. Thanks to this, machines can now more effectively represent and analyze complex data, which has led to significant advancements in various fields such as web search, self-driving cars, and natural language processing. Tasks that were once only doable by humans can now be efficiently accomplished by machines. Recent deep learning methods can match or even surpass humans in task-specific applications, unlike earlier AI algorithms that resulted in subhuman performance [41].

In the healthcare industry, AI is increasingly being used in various applications (i.e., remote patient monitoring, medical diagnostics and imaging, risk management, virtual assistants, etc.). Medical fields that rely on imaging data, such as radiology, pathology, dermatology, and ophthalmology, have already seen benefits from implementing AI tools [41].

AI implementation in the medical field has a key benefit of being able to process and analyze a vast amount of data and identify non-linear correlations between them. These abilities are not achievable by human brains alone and could be incredibly helpful if implemented in the scenario of child abuse and neglect. These can be utilized to perform three basic tasks: (a) prediction, (b) identification/diagnosis, (c) decision process.

(a) Identifying children who are at risk of becoming victims of abuse is a crucial step. Every society should not only diagnose suspected abusive cases promptly but also strive to prevent this phenomenon from occurring in the first place. Taking such preventive measures can help reduce the social burden of child abuse and neglect. To achieve this goal, future research should focus on developing effective AI tools by creating and training AI algorithms with vast amounts of child protective service and healthcare data. Such an approach would enable the implementation of specific interventions aimed at preventing harm against children.

(b) Promptly diagnosing suspected cases of child abuse is crucial yet challenging. The potential to incorporate AI models in the diagnostic process of these events is vast. For instance, AI tools could aid in distinguishing non-accidental fractures from accidental ones. These models could assist radiologists in identifying bone lesions highly indicative of abuse. Furthermore, deep learning algorithms could be utilized to detect new patterns of abuse that have not yet been discovered in bones. AI technologies have the potential to be applied in various fields, including the detection of abusive head trauma. For instance, analyzing retinal changes in patients through AI can aid in distinguishing between abused and non-abused children. In addition, in the future, healthcare professionals may be assisted by AI tools capable of suggesting a suspected diagnosis of child maltreatment by assessing different types of data, such as clinical, imaging, anamnestic information, and more.

(c) After confirming a suspected case of child abuse, it is crucial to identify the most appropriate course of action to ensure the child’s safety. This decision-making process, involving healthcare professionals and child protective services, can include hospitalization, psychological/neuropsychiatric interventions, foster care placement, educational support, and more. These decisions are often complex and non-linear, making it challenging to determine the best course of action. Therefore, AI tools could be instrumental in assisting with these tasks. Thus, it is recommended to conduct further research on this topic to gain better understanding.

### 4.5. Limitations

The first limitation of this study relies on the high heterogeneity of the included manuscripts. In fact, they had several different approaches and designs. Thus, we could not infer specific conclusions from the systematic review. We could only demonstrate the state of the art of AI implementation in the child abuse and neglect field. Second, even if we carefully analyzed the PubMed database, some publications might have been missed if reported in other medical and computer science databases. Indeed, in this interdisciplinary field, eligible publications could also have been reported in non-medical databases. Furthermore, the so-called publication bias could be likely present: several analyses could have not been published due to the low accuracy of the model. Indeed, in this field, only highly accurate algorithms are frequently published. Therefore, the results of many studies may have been missed. The last limitation relies on the choice not to include non-English studies that could have been relevant for understanding the state of AI implementation in the child abuse and neglect field.

## Figures and Tables

**Figure 1 children-10-01659-f001:**
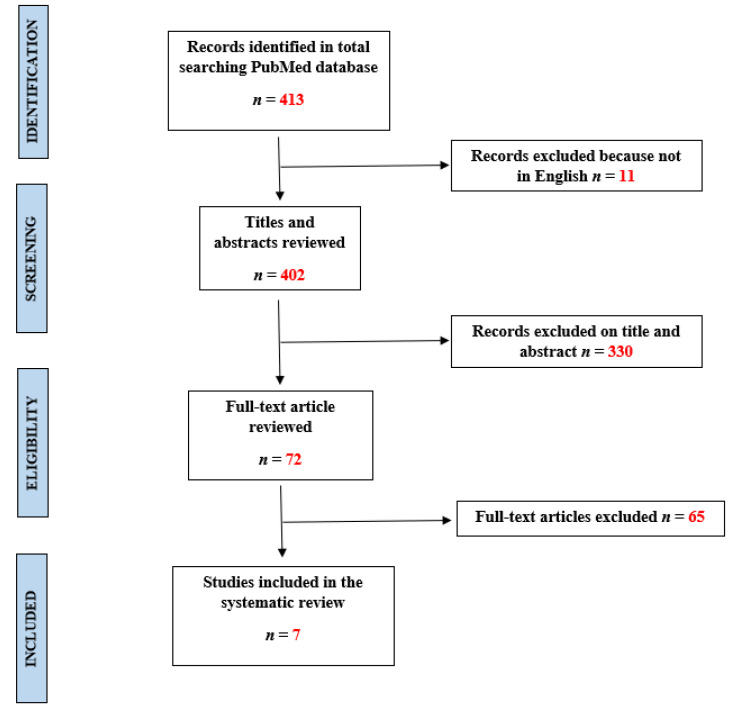
PRISMA flow diagram.

**Table 1 children-10-01659-t001:** Summary of the results.

No	First Author	Year	Input Data	Age/Sex	Type of Abuse	Output/ Prediction	Artificial Intelligence Model	Size of Dataset	Accuracy	Sensitivity	Specificity
1	Tsai A [11]	2022	distal tibia X-rays from skeletal surveys	children ≤ 1 year old; no indications about children’s sex	physical abuse	identify children at low and high risk of abuse	convolutional neural network	normal study cohort: 117 normal distal tibia radiographs from 89 infants; abnormal study cohort: 73 CML images from distal tibia radiographs of 35 infants	Mean 93% (Standard Deviation 1.8%)	Mean 88% (Standard Deviation 5%)	Mean 96% (Standard Deviation 1.5%)
2	Shahi N [12]	2021	demographic and medical data (Model 1 used a combination of demographic and laboratory data; Model 2 used a combination of demographic and laboratory data in association with the text from radiology reports)	children ≤ 18 years old; children of both sexes	physical abuse	identify children who may have been abused	artificial neural network and natural language processing (for Model 1, an artificial neural network; for Model 2, natural language processing)	abusive trauma patients 737; non-abusive trauma patients 575	Model 1 86.3% (95% confidence interval, 84.6–88.0%) Model 2 93.4% (95% confidence interval, 92.2–94.6%)	Model 1 87.2% (95% confidence interval, 85.6–88.8%) Model 2 92.5% (95% confidence interval, 91.2–93.8%)	Model 1 85.1% (95% confidence interval, 83.3–86.9%) Model 2 94.6% (95% confidence interval, 93.5–95.7%)
3	Annapragada AV [13]	2021	unstructured free-text of electronic medical records (including notes from physicians, nurses, and social workers)	no indications about children’s age and sex	physical abuse	identify child abuse from pediatric electronic medical records	three natural language processing algorithms (Bag of Words—BOW; Word Embeddings—WE; Rules-Based—RB)	478 cases positive for physical abuse; 389 cases negative for physical abuse	BOW 89.9% (2.6% Standard Deviation, max 93.1%) WE 65.8% (Standard Deviation 2.8%, max 70.1%) RB 76.6% (3.7% Standard Deviation, max 81.6%)	Not available	Not available
4	Tiyyagura G [14]	2021	text of medical records	children ≤ 1 year old; no indications about children’s sex	physical abuse	identification of high-risk injuries	natural language processing	2000 cases with positive alerts for abuse	Not available	92.7% (95% confidence interval, 79.0–98.1)	98.1% (95% confidence interval, 97.1–98.7)
5	Kissos L [15]	2020	self-figure drawing	no indications about children’s age and sex	sexual abuse	identify self-figure drawings of sexually abused versus non-abused individuals	convolutional neural network	400 self-figure drawings defined as representing abuse; 411 self-figure drawings classified as non-abuse	69%	70%	68%
6	Ucuz I [16]	2020	demographic and clinical data in association with characteristics of the abuse	children in the age range of 5–17 years; children of both sexes	sexual abuse	predict the development of major depressive disorder versus post-traumatic stress disorder after child sexual abuse	artificial neural network	112 for post-traumatic stress disorder Group; 58 for major depressive disorder Group	99.2%	Not available	Not available
7	Marshall DB [17]	2000	child protection system data	no indications about children’s age and sex	all types of abuse	predict children at risk of abuse in child protective service settings	artificial neural network	12,978 child protection system-investigated referrals	79%	72%	Not available

**Table 2 children-10-01659-t002:** Definition of the risk of bias using the PROBAST tool.

No. First Author	Risk of Bias—ROB				Overall ROB
	Participants	Predictors	Outcome	Analysis	
No. 1, Tsai A [11]	+	+	+	−	High
No. 2, Shahi N [12]	+	+	−	−	High
No. 3, Annapragada AV [13]	+	+	+	−	High
No. 4, Tiyyagura G [14]	+	+	−	−	High
No. 5, Kissos L [15]	?	?	+	−	High
No. 6, Ucuz I [16]	+	+	−	−	High
No. 7, Marshall DB [17]	+	+	−	−	High

ROB = risk of bias; + indicates low ROB/low concern regarding applicability; − indicates high ROB/high concern regarding applicability; ? indicates unclear ROB/unclear concern regarding applicability. If ≥1 domain is associated with high concern regarding applicability (−), the overall ROB is high.

## Data Availability

No new data were created or analyzed in this study. Data sharing is not applicable to this article.

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
