# Peer review of "Artificial Intelligence and Child Abuse and Neglect: A Systematic Review"

_children, 2023, doi:10.3390/children10101659_

Round 1

Reviewer 1 Report

Thank you for the opportunity to review this article. Overall I think it offers a really helpful foundation for future studies on AI and child protection, despite the frustratingly low number of studies available for inclusion.

I only have one question:

Line38: Do you mean 'counteract' rather than 'contrast'?

Author Response

Thank you for your comment.

Yes, we mean counteract. We added it in the manuscript.

Reviewer 2 Report

The study itself is hard for a non-scientist to follow but the conclusions seem more than warranted. Your concerns about misdiagnosis is certainly well-founded. Next time you may want to ensure that the research information is more understandable.

The English is ok but there are certain words that even native English speakers are not likely to encounter.

Author Response

Thank you for the comment.

We revised the manuscript trying to solve this issue.

Reviewer 3 Report

Dear Authors,

I was pleased to review your manuscript entitled "Artificial Intelligence and Child Abuse and Neglect: A Systematic Review." The topic is of utmost importance, and your systematic review provides a valuable summary of the current literature in this field. Overall, the paper is well-structured and presents a thorough analysis that will interest the scientific community.

However, there are specific areas where I believe the manuscript could be further strengthened to provide an even more robust and comprehensive contribution. Below are my focused suggestions:

Ethical and Social Implications:

1. Diagnostic Errors. Given the significant impact of diagnostic errors, particularly false positives and false negatives, a more in-depth discussion in this area would be beneficial. The manuscript does not fully delve into the potential ramifications and ethical concerns surrounding these errors, and a more comprehensive treatment would add nuance to your arguments.

2. Informed Consent: It would be prudent to discuss how informed consent is managed for using sensitive medical data to train AI models. As it stands, this critical issue does not receive enough attention in the paper.

Future Directions:

1. Practical Applications. An expanded discussion on how these algorithms could be applied in clinical practice would add great value to the manuscript. This would help the reader understand the translational aspect of your findings, turning it from academic insight to actionable intelligence.

I trust these observations will help enhance the manuscript, and I would be delighted to see a revised version that addresses these points.

Thank you for the opportunity to participate in the review process for this important work. I am available for any further queries or clarification you may require.

Best regards,

Author Response

Thank you for your comment that we fulfilled as requested.

We added "Ethical and Social Implications" from line 321 to line 384:

"4.3 Ethical and Social Implications

It is well established that incidents of child abuse are significantly underreported and underdiagnosed, due to the inability to identify them. Children who have been abused may seek medical assistance, but the abuse can go undetected, leading to further harm, injuries, and even death. Recognizing and diagnosing child abuse at an early stage significantly reduces the risk of recurrence and improves outcomes for the child. It is crucial to acknowledge demonstrated findings of child abuse when they occur. Misinterpreting markers of abuse as normal or as attributable to another diagnosis (false negative) can lead to the child returning to an unsafe environment where they may experience further horrific injury [34,35,36,37].

On the one hand, the scientific community is also aware that false positives can have harmful consequences. Incorrectly diagnosing child abuse can lead to the child not receiving the appropriate care, resulting in legal issues in civil, juvenile, family, divorce, and criminal courts. It can also lead to the wrongful removal of a child from their home and an innocent individual being falsely accused, convicted, and incarcerated. These outcomes can cause immeasurable harm to the children, individuals, and families involved. [34,35,36,37].

It is often overlooked that balancing the risk of a false negative with that of a false positive is a fundamental ethical calculation. Healthcare professionals may be called to choose which outcome (false positive or false negative) is more critical to avoid. To better understand the latter sentence, it can be helpful to consider what happens in another medical field. The United States Preventive Services Task Force made recommendations against routine screening mammography for women aged 40-50. The task force chose to prioritize the harms of false positives (i.e., overdiagnosis, unnecessary anxiety, costs, procedures, and morbidity) over the potential lives that could be saved through screening. This decision ultimately involves a values-based determination of which outcome has higher costs. Similar considerations can be related to the decisions to prioritize false positives or false negatives in child abuse cases.

That said, it is essential to acknowledge that AI tools (especially those based on machine learning techniques) have been described as a black box in which healthcare professionals can only control the input data. AI algorithms and their outputs are often shrouded in complex and opaque reasoning. When faced with such a nebulous black box, it becomes nearly impossible to comprehend or delineate how a machine arrives at its conclusions [38]. This presents a significant ethical issue in the child abuse and neglect field, as using these tools can lead to a lack of control over the inevitable balancing between false positives and false negatives, adding another layer of complexity to an already complex field.

Risk models based on machine learning are typically evaluated using various predictive performance metrics such as precision, recall, etc. However, even when a risk model with 99% precision is developed, it is still prone to errors in 1% of cases. In reality, many clinical machine learning-based risk models do not achieve such high predictive performance. These errors can have significant social and emotional consequences, especially in the domain of child abuse and neglect, where families found to be suspicious of abuse and neglect could face severe outcomes [39]. Furthermore, the performance evaluation metrics for traditional machine learning-based risk models assume that false positives and false negatives carry equal weight. However, it has not been established or validated that this is true for child abuse and neglect cases [39].

To minimize the abovementioned issues, all researchers should always guarantee strict compliance with scientific indications (such as the ones proposed by the PROBAST tool) about how an AI predictive model should be created. This strict compliance should be mandatory to minimize the negative scenarios these predictive tools may generate if not appropriately designed. Indeed, the need to identify new predictive tools to counteract child abuse and neglect does not justify using shortcuts in developing AI algorithms. As suggested by this review, an appropriate choice of sample size, validation, and management of overfitting, optimism, and missing data should always be carried out.

The integration of AI in clinical practice has raised concerns about patient privacy and data sharing. AI systems rely on vast amounts of medical information to provide accurate and reliable responses. However, this concentration of data poses valid concerns about the security and privacy of protected healthcare information, given the potential for unauthorized breaches. Past incidents of unauthorized access to healthcare databases highlight this potential vulnerability. Patients may also be hesitant to consent to the use of private data for training AI, given the possibility of external systems accessing their information without their knowledge. As technology continues to advance, the healthcare industry must take steps to ensure patient data is safeguarded [40]".

We expanded the "Future Perspective" section from line 386 to line 431:

"4.3. Future perspectives

In recent years, AI has made remarkable progress in perception, the process of interpreting sensory information. Thanks to this, machines can now more effectively represent and analyze complex data, which has led to significant advancements in various fields such as web search, self-driving cars, and natural language processing. Tasks that were once only doable by humans can now be efficiently accomplished by machines. Recent deep learning methods can match or even surpass humans in task-specific applications, unlike earlier AI algorithms that resulted in subhuman performance [41].

In the healthcare industry, AI is increasingly being used in various applications (i.e., remote patient monitoring, medical diagnostics and imaging, risk management, virtual assistants, etc.). Medical fields that rely on imaging data, such as radiology, pathology, dermatology, and ophthalmology, have already seen benefits from implementing AI tools [41].

AI implementation in the medical field has a key benefit of being able to process and analyze a vast amount of data, and identify non-linear correlations between them. These abilities are not achievable by human brains alone and could be incredibly helpful if implemented in the scenario of child abuse and neglect. These can be utilized to perform three basic tasks: a) prediction, b) identification/diagnosis, c) decision process.

a) Identifying children who are at risk of becoming victims of abuse is a crucial step. Every society should not only diagnose suspected abusive cases promptly but also strive to prevent this phenomenon from occurring in the first place. Taking such preventive measures can help reduce the social burden of child abuse and neglect. To achieve this goal, future research should focus on developing effective AI tools by creating and training AI algorithms with vast amounts of child protective service and healthcare data. Such an approach would enable the implementation of specific interventions aimed at preventing harm against children.

b) Promptly diagnosing suspected cases of child abuse is crucial, yet challenging. The potential to incorporate AI models in the diagnostic process for these events is vast. For instance, AI tools could aid in distinguishing non-accidental fractures from accidental ones. These models could assist radiologists in identifying bone lesions highly indicative of abuse. Furthermore, deep learning algorithms could be utilized to detect new patterns of abuse that have not yet been discovered in bones. AI technologies have the potential to be applied in various fields, including the detection of abusive head trauma. For instance, analyzing retinal changes in patients through AI can aid in distinguishing between abused and non-abused children. In addition, in the future, healthcare professionals may be assisted by AI tools capable of suggesting a suspected diagnosis of child maltreatment by assessing different types of data, such as clinical, imaging, anamnestic information, and more.

c) After confirming a suspected case of child abuse, it is crucial to identify the most appropriate course of action to ensure the child's safety. This decision-making process, involving healthcare professionals and child protective services, can include hospitalization, psychological/neuropsychiatric interventions, foster care placement, educational support, and more. These decisions are often complex and non-linear, making it challenging to determine the best course of action. Therefore, AI tools could be instrumental in assisting with these tasks. Thus, it is recommended to conduct further research on this topic to gain a better understanding".

Thanks to your suggestions we strengthened our analysis.

Reviewer 4 Report

This is a useful and interesting article. Its value lies in the methodologies explored, rather than in any any substantive findings. The article explores AI methodologies in monitoring and reporting cases of child abuse and neglect which may be applied to future research studies. As such, the results of this article may be referenced and integrated into novel attempts to apply AI research technology in fresh studies, including valuable longitudinal studies that can identify useful intervention strategies. The most valuable part of the article is the discussion Section 4, from which researchers may draw inferences and guidelines for future studies. 

Some minor points. As a translator of medical literature from German into English, I regret that studies that were not in English were excluded - why so? I'm also aware of some useful medical studies in Mandarin, with an English abstract. Could the authors of the article under review say definitively that the  non-English studies identified were not relevant for their review?

The "median number" of cases for the the included studies is given as "2,559.6". This could be rounded up to  "2,560", surely?

Author Response

Thank you for the comment.

We reported a median size of 2,599.6 cases. We rounded it adding 2,600.

Sadly, we did not have the tools (nor the funds) to include also non-English studies. We agree with the reviewer. We cannot say that the  non-English studies were not relevant. Thus, we added the following statement at the end of the "Limitation" section (line 444):

"The last limitation relies on the choice not to include non-English studies that could have been relevant for understanding the state of AI implementation in the child abuse and neglect field".